# Illuminating T cell-dendritic cell interactions in vivo by FlAsHing antigens

Munir Akkaya[1,2,3†], Jafar Al Souz[4†], Daniel Williams[4], Rahul Kamdar[4], Olena Kamenyeva[4], Juraj Kabat[4], Ethan Shevach[4], Billur Akkaya[3,5*]

[1]Department of Internal Medicine, Division of Rheumatology and Immunology, The College of Medicine, The Ohio State University, Columbus, United States; [2]Microbial Infection and Immunity, The Ohio State University Wexner Medical Center, Columbus, United States; [3]Pelotonia Institute for Immuno-Oncology, The Ohio State University, Columbus, United States; [4]National Institute of Allergy and Infectious Diseases, Bethesda, United States; [5]Department of Neurology, The Ohio State University Wexner Medical Center, Columbus, United States

**\*For correspondence:**
billur.akkaya@osumc.edu

[†]These authors contributed equally to this work

**Competing interest:** The authors declare that no competing interests exist.

**Abstract** Delineating the complex network of interactions between antigen-specific T cells and antigen presenting cells (APCs) is crucial for effective precision therapies against cancer, chronic infections, and autoimmunity. However, the existing arsenal for examining antigen-specific T cell interactions is restricted to a select few antigen-T cell receptor pairs, with limited in situ utility. This lack of versatility is largely due to the disruptive effects of reagents on the immune synapse, which hinder real-time monitoring of antigen-specific interactions. To address this limitation, we have developed a novel and versatile immune monitoring strategy by adding a short cysteine-rich tag to antigenic peptides that emits fluorescence upon binding to thiol-reactive biarsenical hairpin compounds. Our findings demonstrate the specificity and durability of the novel antigen-targeting probes during dynamic immune monitoring in vitro and in vivo. This strategy opens new avenues for biological validation of T-cell receptors with newly identified epitopes by revealing the behavior of previously unrecognized antigen-receptor pairs, expanding our understanding of T cell responses.

## eLife assessment

This is an **important** study that develops a method to fluorescently label peptide MHC complexes on live dendritic cells to enable detection of antigen specific T cells in polyclonal populations. **Solid** evidence that this can be used to effectively identify antigen specific T cells in vitro and in vivo is provided for one model antigen systems (Ova-OTII). The approach has exciting potential as prior single step methods with directly conjugated single peptides have generally failed due to high background. Thus, this approach potentially moves the state of the art forward, but further work is needed to realise and determine the limits and ultimate utility of the approach.

## Introduction

The interactions between T cells and APCs form the cornerstone of adaptive immunity. Within the intricate milieu of lymphoid tissue, T cell priming stands out as a discrete and precise process, occurring amidst the multitude of dynamic cellular interactions. Multimerized peptide-major histocompatibility complex (pMHC) labeling has been the gold standard tool for capturing antigen-specific T cells from a polyclonal repertoire (*Jenkins et al., 2010*). However, such multimers, most commonly tetramers, have limitations in monitoring dynamic interactions of antigen-specific T cells in real-time due to their blocking effect on the immune synapse. Likewise, currently available T cell receptor-like

(TCRL), also known as peptide-in-groove (PIG), antibodies that detect pMHC complexes are known to disrupt the immune synapse, making them unsuitable for real-time monitoring of antigen-specific T cell interactions (*Dahan and Reiter, 2012*; *Høydahl et al., 2019*). Although alternative approaches for labeling antigenic peptides with conventional tags such as fluorescent proteins, fluorophores, and biotin-streptavidin have been used, these tags have limitations such as size constraints, membrane impermeability, and photophysically unstable nature, which restrict their use in applications for dynamic monitoring (*Giepmans et al., 2006*; *Hoffmann et al., 2010*; *Toseland, 2013*; *Liu and Cui, 2020*).

To overcome these drawbacks, we have developed a highly specific and minimally disruptive labeling strategy for studying antigen-specific T cell-APC interactions. Our approach involves introducing a six-amino acid tetracysteine tag (CCPGCC) on antigenic peptides and inducing fluorescence using membrane permeable thiol-reactive Arsenical Hairpin (AsH) probes (*Hoffmann et al., 2010*; *Adams et al., 2002*). We have engineered the OVA$_{(323-339)}$ peptide by placing the tetracysteine tag at either the amino or the carboxy terminus, away from the TCR and MHC binding sites, to avoid disrupting T cell- APC synapse in vitro and in vivo (*Robertson et al., 2000*). Owing to these modifications, our labeling strategy preserves the T cell priming ability and allows for the detection of the fluorescent signal on the surface of APCs in an MHCII-dependent manner. Using flow cytometry and live microscopy, we found that the labeled peptide is readily taken up by the cognate CD4$^+$ T cells with precise specificity in vivo, as previously shown for pMHC complexes taken up from the immune synapse via processes such as trogocytosis and transendocytosis (*Aucher et al., 2008*; *Daubeuf et al., 2006*; *Wetzel et al., 2005*; *Hudrisier et al., 2001*; *Huang et al., 1999*). Furthermore, we have created new variants of the tetracysteine-tagged OVA$_{(323-339)}$ peptide with altered TCR-binding sites and have shown that the AsH probe labeling can track the TCR-mediated uptake of nuanced pMHC complexes. This indicates that our labeling strategy is highly robust and forges a novel path for investigating immune responses against altered antigens such as those derived from self, microbiota, and tumor microenvironment. Overall, AsH probe-mediated immune monitoring offers significant advantages over existing methods, making it a versatile tool for in vitro and in vivo applications including live imaging of the T cell synapse within intact tissue architecture. This novel approach will further our understanding of T cell-mediated immune responses by also complementing tetramers for sorting antigen-specific T cells with precision and extending it to newly discovered epitopes.

## Results

### Signal intensity is determined by the sequence and placement of tetracysteine tag

We first generated OVA$_{323-339}$ variants by placing a tetracysteine tag 'CCPGCC' at amino or carboxy termini, either consecutively or via a flexible $\epsilon$-amino-caproic acid ($\epsilon$-ACA) linker to prevent endosomal-lysosomal enzymatic digestion to the tag. This yielded four variant peptides named as C- (carboxy terminus without linker), N- (amino terminus without linker), NACA- (amino terminus with $\epsilon$-ACA), and CACA- (carboxy terminus with $\epsilon$-ACA) (*Figure 1a*). We used these variants to load mature C57BL/6 CD11c$^+$ splenic DCs from C57BL/6 mice and induced fluorescence by treating cells with Fluorescein-AsH (FlAsH) probe. Flow cytometry analysis showed that CACA- peptide combination but not the other tested variants induced a distinguished fluorescent signal (*Figure 1b*). Cysteine is naturally oxidation prone and this may reduce FlAsH binding to tetracysteine tag. We addressed this problem by pre-treating OVA$_{CACA}$ with tris-2-carboxyethylphosphine (TCEP), an irreversible reducing agent that blocks cystine disulfide bridge formation and elicited a consistent FlAsH signal (*Figure 1c*). Moreover, we used another AsH probe excitable by 561 nm laser, called ReAsH, to occupy non-specific cysteine residues that would otherwise attract FlAsH (*Stroffekova et al., 2001*). Pre-incubation of DCs with ReAsH improved signal-to-noise ratio significantly (*Figure 1d*). We have created additional OVA variants to enhance signal intensity. These variants include one with two tetracysteine residues and another with a longer tetracysteine tag that withstands the British anti-Lewisite (BAL) wash, a crucial step to reduce background labeling by removing excess AsH probe (*Machleidt et al., 2007*). However, new variants provided FlAsH intensity comparable to the original CACA variant, indicating no additional benefit for longer tags (*Figure 1e*).

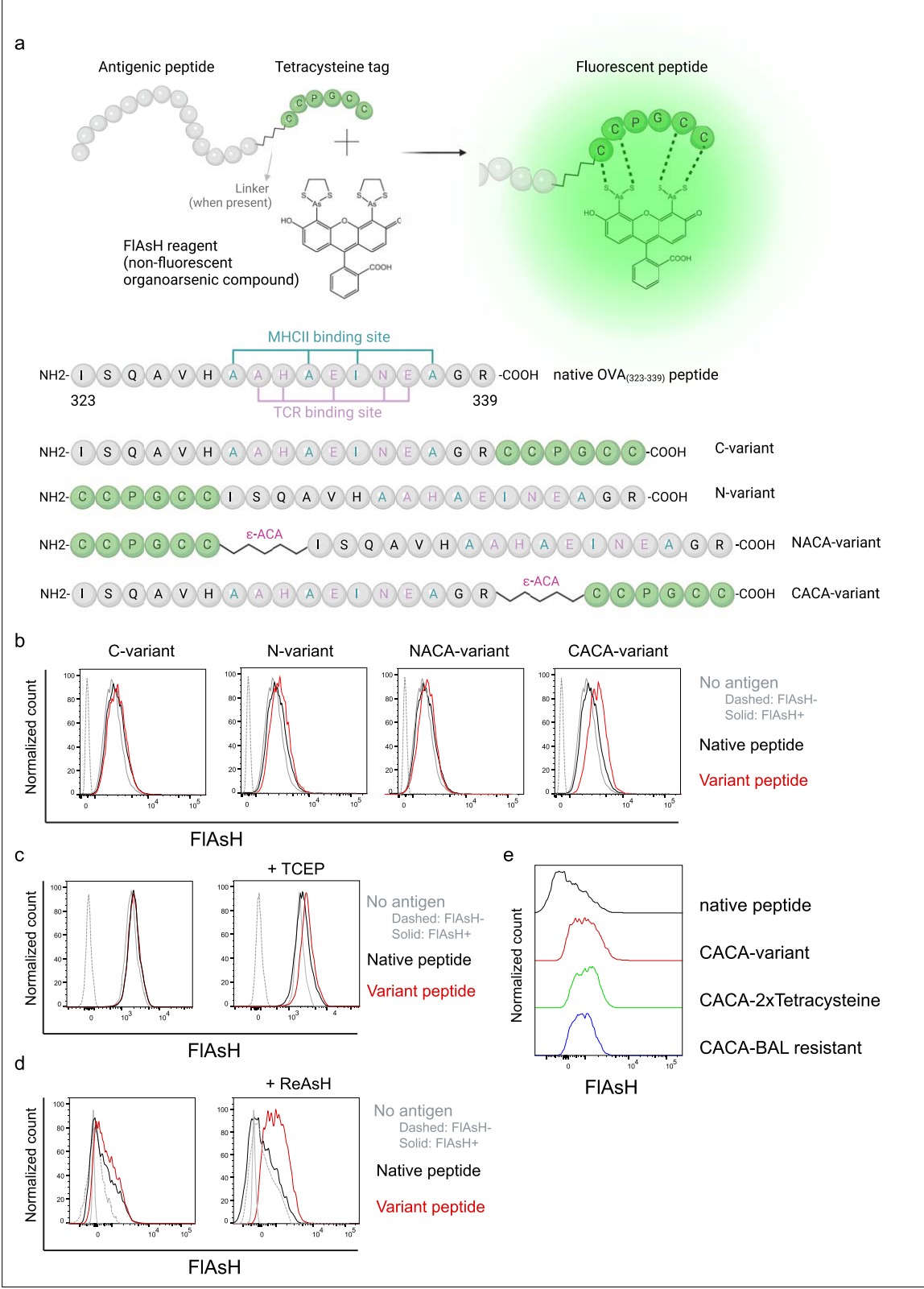

**Figure 1.** Fluorescein-AsH (FlAsH) labeling elicits fluorescence depending on the location of the tetracysteine tag. (**a**) Arsenical Hairpin (AsH)- probe labeling strategy for the tetracysteine-tagged peptides and the four OVA$_{323-339}$ variants generated for the study. (**b**) FlAsH signal intensity on splenic dendritic cells (DCs) pulsed with tagged OVA$_{323-339}$ variants. (**c**) FlAsH signal intensity on splenic DCs pulsed with CACA-variant following treatment with Tris Carboxy Ethyl Phosphene (TCEP), a strong reducing agent. (**d**) Signal intensity following pre-treatment of splenic DCs with ReAsH, followed

*Figure 1 continued on next page*

*Figure 1 continued*

by pulsing with CACA-variant and FlAsH treatment. (**e**) FlAsH signal intensity of modified CACA-variant peptide with a duplicated CCPGCC sequence (CACA-2xTetracysteine) and CACA-variant peptide flanked with BAL-resistant peptide sequences (CACA-BAL resistant). Data in (**b–d**) are representative of three independent experiments, each performed with three biological replicates.

## Tetracysteine tag does not alter the immunogenicity of peptide

Next, we sought to compare the CD4$^+$ T cell priming ability of four tetracysteine-tagged OVA$_{323-339}$ variants and that of native peptides in vitro. To do so, naïve CD4$^+$ T cells were isolated from OVA$_{323-339}$ specific TCR transgenic OTII mice, labeled with a proliferation dye, and cocultured with mature CD11c$^+$ DCs that were pre-loaded with equimolar amount peptides for three days. Proliferation and activation status of OTII T cells were not altered by the presence or placement of the tag (*Figure 2a–d*). Subsequently, DCs were pulsed with OVA$_{CACA}$ vs. native peptide, treated with FlAsH, and cocultured with fluorescently labeled OTII T cells for live confocal microscopy imaging. Within the first hours of coculture, OTII T cells established stable contact with OVA$_{CACA}$-pulsed DCs, and FlAsH-labeled peptide gradually concentrated at the contact site (*Figure 2e*). At 24 hr after treatment, OTII T cells in the peptide-treated groups exhibited increased volume and a shift towards a less spherical shape, indicative of T cell blasting (*Figure 2f*). Furthermore, the interaction of T cells with OVA$_{CACA}$ and native OVA$_{323-339}$ pulsed DCs resulted in a comparable decrease in T cell motility, suggesting that the tag had no impact on their dynamics in vitro (*Figure 2g*).

Next, we performed a series of adoptive transfers to measure the immunogenicity and signal persistence of tetracysteine-tagged peptides in vivo using two-photon microscopy and flow cytometry. Splenic DCs from β-Actin DsRed mice were treated with ReAsH to prime them for FlAsH labeling. Because DsRed and ReAsH do not get transferred to T cells during immune synapse, their overlapping fluorescence emission do not pose any problem for tracking the DCs. Next, DCs were pulsed with a peptide mix comprised of equimolar amounts of OVA-Biotin and OVA$_{CACA}$, where OVA-Biotin served as a positive control for flow cytometry. This was followed by optimized FlAsH treatments as indicated in *Figure 1* and adoptive transfer into C57BL/6 recipients via footpad injection. Naïve CD4$^+$ T cells were isolated from OTII and WT C57BL/6 spleens, labeled with distinct cell-tracker dyes, mixed 1:1, and transferred into recipients via retroorbital i.v. injection (*Figure 3a*). As previously demonstrated, adoptively transferred DCs were detectable in the popliteal lymph node 18–24 hr post-transfer and they largely disappeared by 48 hr (*Bajénoff et al., 2003*; *Bousso, 2008*; *Ingulli et al., 1997*). FlAsH signal remained stable on DCs for the entire duration DCs were present in the lymph node (*Figure 3b*). Strikingly, FlAsH was picked up by the cognate OTII T cells, suggestive of an antigen-specific peptide major histocompatibility complex II (pMHCII) trogocytosis in-vivo (*Figure 3c and d*). Antigen-specific transfer was also confirmed by OVA-Biotin staining on OTII T cells by fluorochrome-conjugated streptavidin (*Figure 3c and d*). We had previously shown that antigen-specific Treg cells trogocytose pMHCII from DC synapse more efficiently than their effector CD4 +T cell counterparts (*Akkaya et al., 2019*). To quantify FlAsH signal picked up by OVA$_{323-339}$-specific Treg cells, naïve OTII T cells were differentiated into induced Treg (iTreg) cells. While iTreg cells cocultured with antigen-pulsed DCs performed greater MHCII trogocytosis than naïve T cells at 18 hr in vitro, peptide-specific trogocytosis was indiscernible within this time frame (*Figure 2—figure supplement 1a, b*). Similar to naïve OTII T cells, OTII iTreg cells acquired OVA$_{CACA}$ and OVA-Biotin, with distinct peptide-specific signal and proliferative response on day 2 post-adoptive transfer (*Figure 2—figure supplement 1c–f*).

Intravital two-photon microscopy of the popliteal lymph node further confirmed the antigenicity of OVA$_{CACA}$ in vivo by revealing the morphological changes compatible with T cell priming such as increased T cell volume and decreased sphericity (*Figure 3e*). OTII-DC synapses had greater 3D volume and duration and harbored more FlAsH than surrounding regions (*Figure 3g–h*). Overall, FlAsH signal was exclusively picked up by OTII T cells imaged, as early as 18 hr (*Figure 3i*, *Video 1*).

## AsH-probe robustly tracks pMHCII complexes

To delineate whether FlAsH labeling reports free vs. MHCII-bound peptide in cells, we initially assessed how surface MHCII expression of cells correlates with the OVA$_{CACA}$-FlAsH signal. C57BL/6 splenocytes were incubated with OVA-Biotin or OVA$_{CACA}$ and peptide distributions among the T cells, B cells and DCs were quantified by flow cytometry using streptavidin or FlAsH (*Figure 4a*). Both streptavidin and FlAsH signals correlated significantly with MHCII expression, providing the strongest signal in

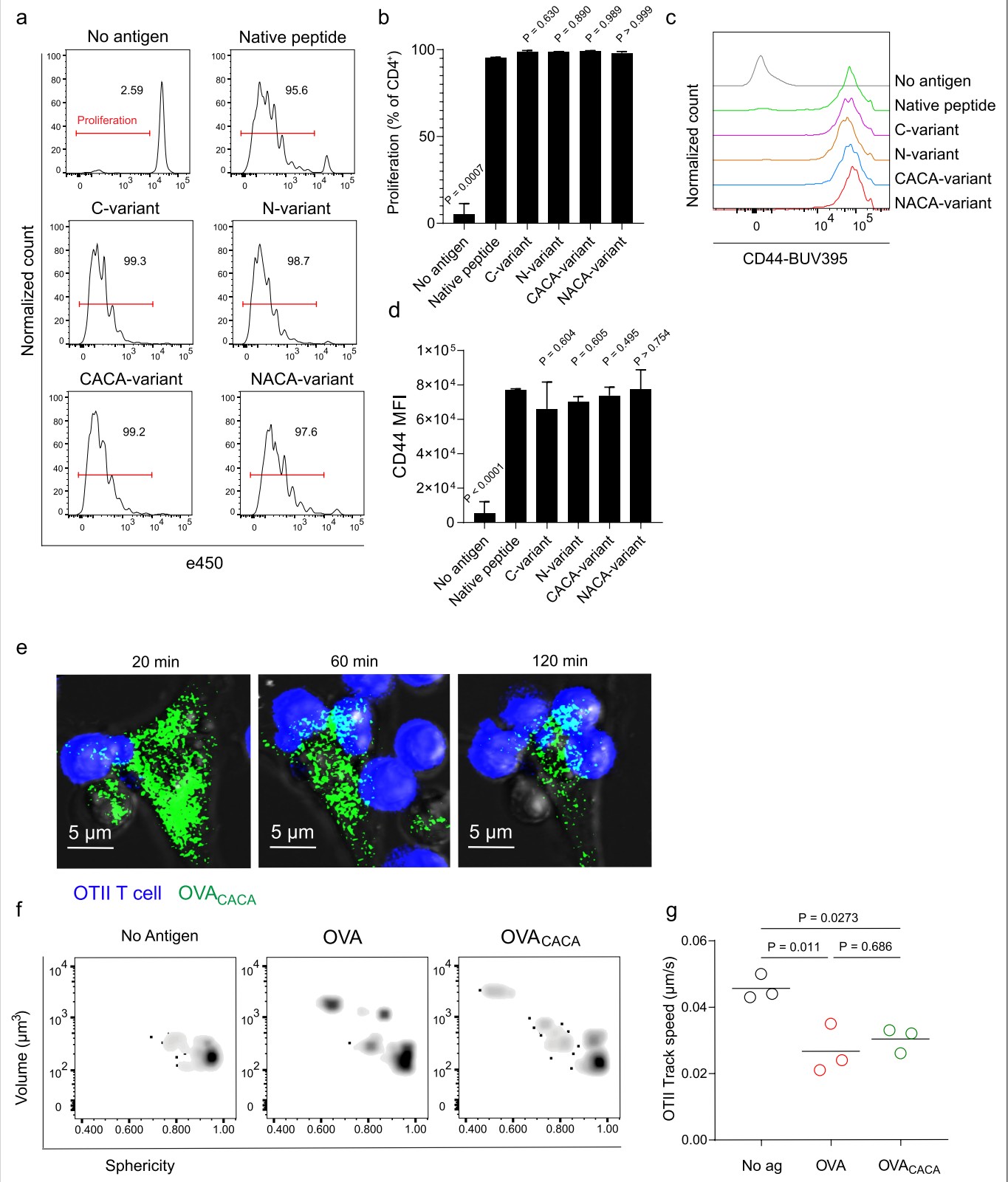

**Figure 2.** Tetracysteine tag does not alter the T cell priming ability of the native peptide. (**a–d**) Dendritic cells (DCs) were loaded with the indicated peptides (5 µM) and co-cultured with e450-labeled naïve OT-II T cells (5 × 10⁴) at a 1:10 DC to T cell ratio. (**a**) Histograms (**a**) and the bar graph (**b**) demonstrate the proliferation status of T cells at 72 hr. Histograms (**c**) and the bar graph (**d**) depict the surface expression of CD44, a T cell activation marker. The bars (**b,d**) indicate the mean of three biological replicates; the error bars show the standard deviation. Data are representative of three

*Figure 2 continued on next page*

*Figure 2 continued*

independent experiments. p-values refer to comparisons between each group and the native peptide, calculated using one-way ANOVA with Dunnett's multiple comparisons test. (**e–g**) Peptide-pulsed DCs ($2 \times 10^5$) were treated with Fluorescein-AsH (FlAsH), cocultured 1:1 with e450-labeled naïve OT-II T cells and imaged for real-time interactions for 3 hr (blue, OTII; green, FlAsH). (**e**) Time-lapse images depict the clustering of OTII T cells around OVA$_{CACA}$-pulsed DCs. (**f**) Histocytometry plots for OTII T cell volume and sphericity. (**g**) Graph shows the average track speed of OTII T cells. The lines mark the means of n=3 replicates (one per symbol). Data are representative of two independent experiments. p-values were calculated using a one-way ANOVA with Tukey's test.

The online version of this article includes the following figure supplement(s) for figure 2:

**Figure supplement 1.** OVA$_{323-339}$-specific Treg cells acquire OVA$_{CACA}$-Fluorescein-AsH (FlAsH) from immune synapse.

DCs, where CD4$^+$ T cells served as a negative control due to lack of MHCII expression in mouse T cells (*Schooten et al., 2005*; *Figure 4b*, *Figure 4—figure supplement 1*). It is worth mentioning that streptavidin detection of biotinylated peptide is a viable option for quantifying peptide loading on cells. However, size and membrane impermeable nature of streptavidin makes it unsuitable for assays that include live monitoring of the T-DC interaction (*Dundas et al., 2013*). Next, we addressed whether FlAsH labeling of OVA$_{CACA}$-pulsed DCs is dependent on the MHC expression. CD11c$^+$ DCs were isolated from MHCII single knockout or MHCI and II double knockout (KO) mice, labeled with tracking dyes, and mixed 1:1 with WT C57BL/6 DCs. Subsequently, they were incubated with OVA$_{CACA}$ and treated with ReAsH and FlAsH as in *Figure 1* (*Figure 4e*). To make a fair comparison between WT and KO DCs and minimize autofluorescence, we performed adoptive transfers, aiming to facilitate in vivo elimination of apoptotic and possibly defective KO DCs (*Humblet-Baron et al., 2019*; *Loschko et al., 2016*). Flow cytometry analysis of the draining lymph node demonstrated that WT DCs, but not MHCII KO DCs, harbored FlAsH. Two-photon microscopy of the lymph node further confirmed these findings, indicating that FlAsH robustly marks OVA$_{CACA}$-MHCII complexes (*Figure 4e–h*).

To address how FlAsH acquisition from synapse marks antigen-specific vs. bystander T cells, DCs were labeled with ReAsH and cell membrane dye PKH-26, pulsed with OVA$_{CACA}$, Eα$_{(52-68)}$ or left unpulsed. Next, they were treated with FlAsH and injected into the recipient mice via footpad. Subsequently, 1:1 mixture of labeled naïve OTII and Eα-specific T cells were adoptively transferred into via retroorbital sinus i.v. and the popliteal lymph nodes of the recipients were removed 36–42 hr after the transfer (*Figure 5a*). Flow cytometry analysis showed that OTII and Eα-specific T cells both acquired PKH-26 from double-pulsed DCs, indicating that they captured DC cell membrane from synapse (*Figure 5b*). This process is termed as TCR-mediated trogocytosis and has previously been described as a feature of the DC-T cell synapse (*Daubeuf et al., 2006*). Although both T cell types trogocytosed double-pulsed DC cell membrane, FlAsH signal was picked up only by OTII T cells, suggesting that FlAsH tracks pMHCII mobility with a high precision (*Figure 5c*).

Furthermore, we visualized dynamic CD4$^+$ T cell-DC contact and pMHCII mobility in vivo at 18 hr after the transfer. DCs were labeled with ReAsH and pulsed with OVA$_{CACA}$ and/or LCMV GP$_{(61-80)}$ and treated with FlAsH. Recipient mice were injected with unpulsed, OVA$_{CACA}$ single or OVA$_{CACA}$- GP$_{(61-80)}$ double pulsed DCs via footpad along with a 1:1 mixture of labeled naïve OTII and SMARTA T cells i.v. via retroorbital sinus. Popliteal lymph node microscopy confirmed that OTII and SMARTA T cells had stable contacts with double-pulsed DCs (*Figure 5d*, *Video 2*). Similar to Eα-specific T cells, SMARTA also failed to pick up FlAsH signal despite showing signs of antigen-specific contact such as increased volume, reduced sphericity, and decreased track speed (*Figure 5e–f*).

## Tetracysteine-tagging of peptides provides a sensitive tool to visualize interactions with altered TCR affinity

The lack of adaptable antigen-targeted probes presents a significant obstacle in the dynamic visualization of antigen-specific immunity. To address this gap, we sought to determine the robustness of tetracysteine tagging in visualizing nuanced pMHCII variants. We created three variants by introducing alanine mutations at the TCR binding residues 331, 333, and 335 of OVA$_{CACA}$ (*Figure 6a*). The mutations at 331 and 333, depicted as variants 1 and 2, respectively, were previously reported to abolish the binding of OTII TCR to OVA$_{323-339}$ (*Robertson et al., 2000*). Although the residue 335 is flanked between the core OTII TCR binding residues 333 and 336, mutant 335 was shown to elicit the normal proliferative response in vitro (*Robertson et al., 2000*). We addressed how pMHCII uptake by T cells is influenced by the mutation of TCR binding residues of OVA$_{CACA}$. DCs were isolated from CD45.1

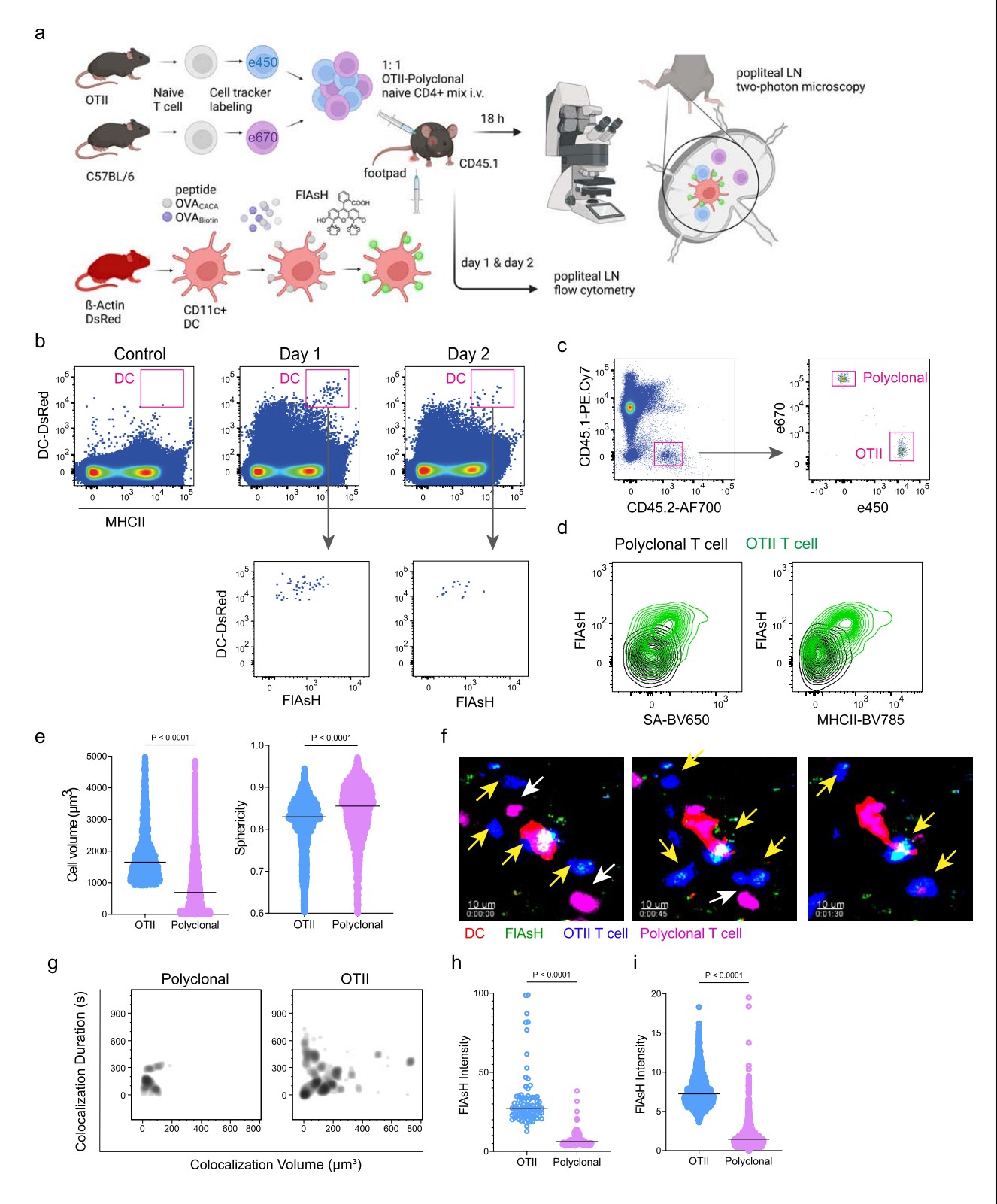

**Figure 3.** T cells acquire tetracysteine-tagged peptide from immune synapse. DsRed+ splenic dendritic cells (DCs) were double-pulsed with 5 μM OVA_CACA and 5 μM OVA-biotin and adoptively transferred into CD45.1 recipients (2×10^6 cells) via footpad. Naïve e450-labeled OTII (1×10^6 cells) and e670-labeled polyclonal T cells (1×10^6 cells) were injected i.v. Popliteal lymph nodes were either imaged at 18 hr or analyzed by flow cytometry at 18 hr or 48 hr post-transfer. (**a**), Experimental approach. (**b**) Flow cytometry plots gating on DsRed+ DCs over 48 hr (top panel) and demonstrating the

*Figure 3 continued on next page*

Figure 3 continued

OVA_CACA levels (Fluorescein-AsH: FlAsH, bottom panel). (**c**) Gating strategy for T cells of the recipient (CD45.1) and donor (CD45.2). (**d**) T cell levels of OVA_CACA (FlAsH), MHCII, and OVA-biotin (detected with streptavidin) at 48 hr following adoptive transfer. (**a–d**) Data are representative of three independent experiments with n=3 mice per time point. (**e–i**) Summary data from live microscopy showing OTII cell blasting (**e**), DC–T cell interaction with 45 min intervals (**f**), plots for the DC-T cell contact duration and volume, graphs for the FlAsH intensity limited to the DC-T cell contact regions (**g–h**) and average Flash intensity acquired by T cells (**i**). Data are representative of four independent experiments with n=2 mice per experiment. p-values were calculated using two-sided Welch's t-test.

congenic mice, loaded with variant peptides, and cocultured with naïve CD45.2 OTII T cells for three days (*Figure 6b–e*). In line with earlier observations, variants 1 and 2 interrupted T cell priming while variant 3 led to a proliferative response similar to native peptide (*Figure 6c–d*). TCR-mediated uptake of the FlAsH-labeled pMHCII complexes followed the same pattern and only variant 3 complexes produced a comparable signal to that of OVA_CACA (*Figure 6e*). These suggest that AsH-probe labeling can be a sensitive method to track low affinity/nuanced pMHCII complexes.

## Discussion

Dynamic monitoring of antigen-specific T cell responses in vivo poses challenges due to the limited availability of reagents and the size constraints of the immune synapse. Here, we address this issue by presenting a novel strategy for visualizing T cell-APC interactions via tetracysteine tagging of antigenic peptides. While tetracysteine tagging has been previously applied in cell biology and virology, these applications have been mostly limited to cell lines and in vitro studies in immunology-unrelated experimental settings (*Griffin et al., 1998*; *Mohl and Roy, 2017*; *Li et al., 2022*; *Ng et al., 2019*). On the other hand, our technique labels antigenic peptides and tracks pMHCII complexes in primary APCs during their interactions with T cells both in vitro and in vivo. Our approach is superior as compared to other strategies for monitoring T cell-DC interactions, in the way it robustly targets specific pMHCII complexes in real-time in vivo. There is no need for special organisms or genetic modifications, and the technique can easily be expanded to new antigens and T cell-APC pairs from both mice and humans. This strategy can, for instance, be integrated into validation efforts for antigen discovery and utilized in the investigation of the synapses of newly deorphanized human T cells, a largely unexplored area where tools are scarce (*Dobson et al., 2022*; *Wang et al., 2022*; *Lewis and Peters, 2023*; *Pogorelyy et al., 2022*). Given that is a two-part system (tag and FlAsH), it also carries the added capability of being inducible at different time points to allow for increased flexibility in interrogating pMHCII-TCR interactions over time. On the other hand, this introduces a high background due to the non-specific binding of FlAsH to endogenous cysteines. Thus, we implemented a ReAsH pretreatment strategy, successfully improving the signal-to-noise profile for FlAsH. It's worth mentioning our unpublished attempts to load dendritic cells (DCs) with preconjugated peptide and FlAsH to mitigate the background. Unfortunately, this endeavor led to almost a complete loss of the FlAsH signal and has not been pursued further. Therefore,

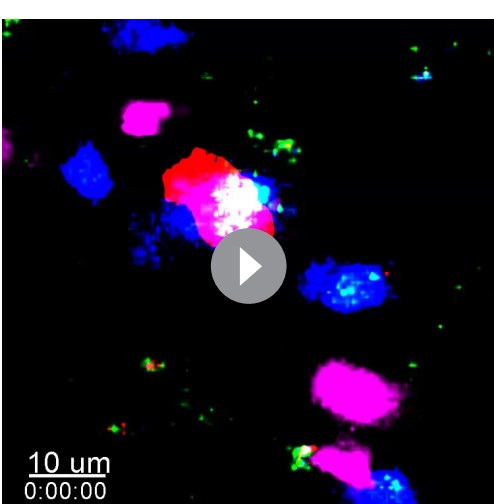

**Video 1.** OVA_CACA-pulsed DsRed+ dendritic cells (DCs) were adoptively transferred via footpad. Naïve e450-labeled OTII and e670-labeled polyclonal T cells (1 × 10$^6$ cells) were injected i.v. via retroorbital sinus. Popliteal lymph nodes were imaged by intravital two-photon microscopy at 18 hr after adoptive transfer. Video shows movement and interaction of naive OT-II (blue) cells with DCs (red). OVA_CACA-Fluorescein-AsH, FlAsH is visualized as cyan-green speckles in DCs and OTII cells. Polyclonal T cells (magenta) don't engage in sustained interactions with DCs and don't acquire FlAsH. Data are representative of four independent experiments.

https://elifesciences.org/articles/91809/figures#video1

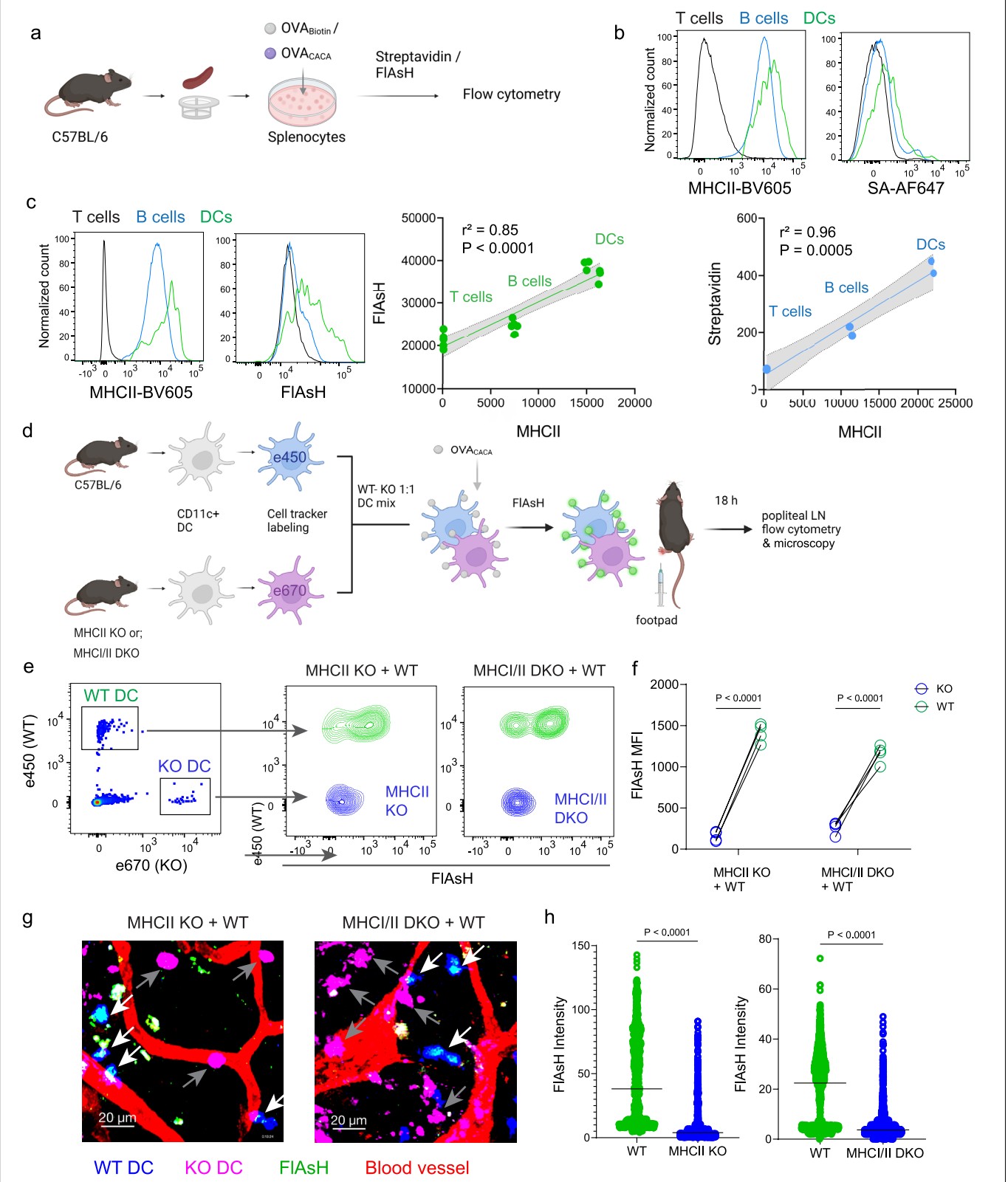

**Figure 4.** Arsenical Hairpin (AsH) probe reports the peptide-loaded on the MHCII. (**a-c**), Relationship between surface MHCII expression and peptide load. (**a**) Experimental scheme. (**b**) Plots (top) show the surface levels of MHCII and OVA-biotin, graph (bottom) shows their correlation. Data are representative of two independent experiments performed with n=2 biological replicates (one per symbol). (**c**) Surface levels of MHCII and $OVA_{CACA}$ (Fluorescein-AsH, FlAsH) and their correlation. Data are representative of two independent experiments performed with n=6 biological replicates (one

*Figure 4 continued on next page*

*Figure 4 continued*

per symbol). Simple linear regression and Pearson correlation were used to demonstrate the relationship between surface MHCII and peptide load, gray area and dotted lines mark the standard error of the fitted line (**b, c**). (**d**) Schematic depicting in vivo adoptive transfer approach. Wild-type (WT) and KO DCs were labeled, mixed, pulsed with 5 μM OVA$_{CACA}$, labeled with FlAsH and adoptively transferred to WT recipients via footpad (1–2 × 10$^6$ cells). (**e–f**) Flow cytometry plots (**e**), and graphs (**f**) showing FlAsH intensity of WT, MHCII-KO, or MHCI/II double-KO DCs that had migrated to the lymph node within 18 h following adoptive transfer. Data are representative of two independent experiments with n=4 mice per group. p-values were calculated using two-sided Student's t-test. (**g–h**) Microscopy images of popliteal lymph node following anti-CD31 (red) antibody injection i.v., yellow arrows point to WT, white arrows point to KO DCs (**g**), FlAsH intensity of adoptively transferred dendritic cells (DCs) (**h**). Data are representative of four independent experiments with n=2 mice per experiment. p-values were calculated using two-sided Welch's t-test.

The online version of this article includes the following figure supplement(s) for figure 4:

**Figure supplement 1.** Sequential gating strategy for splenocytes to analyze CD4+ T cells, B cells, and dendritic cells (DCs).

exploring new peptide probes chemically conjugated to small organic dyes emerges as a potential avenue for future investigation.

Most important of all, our experiments demonstrate that tetracysteine-tagged peptides can be transferred from DCs to T cells via synaptic processes, such as antigen-specific pMHCII trogocytosis/transendocytosis, marking acceptor T cells for several days. Although the turnover of pMHCII in acceptor T cells and its temporal relationship to the TCR downstream events require further elucidation, our visualization of tetracysteine-tagged pMHCII can complement tetramers in detecting antigen-specific T cells in vivo. Additionally, we show that tetracysteine tagging of pMHCII does not broadly alter or disturb T cell-DC synapse by functional read-outs. Although the ratio of peptide-FlAsH conjugates to free peptide in DCs would be needed to support this claim, our approach offers an advantage over tetramers in that it is suitable for real-time monitoring of the immune synapse. Finally, it is worth noting that tetracysteine tags can be visualized as electron-dense particles by Transmission Electron Microscopy (*Ellisman et al., 2012*; *Cortese et al., 2009*), providing potential for the detailed investigation of T-DC synapse and subcellular localization of pMHCII complexes. Although TEM application was out of scope for this study, it offers a promising avenue for future research. In conclusion, the tetracysteine labeling approach described here provides a powerful and unique tool to advance our understanding of T cell-APC interactions, offering new opportunities to investigate the immune response and develop improved immunotherapies.

## Materials and methods
### Animals and reagents

C57BL/6, DsRed.T3, *H2-K1/H2-D1* knockout (MHCI KO), Eα$_{52-68}$, LCMV GP$_{61–80}$-specific (SMARTA), and OVA$_{323–339}$-specific (OT-II) TCR transgenic mice were acquired from The Jackson Laboratory. H2-Ab1 knockout (MHCII KO) mice were obtained from Taconic Biosciences. MHCI-II double knockout mice were obtained by breeding single knockout mice to homozygosity. All mice were maintained in Ohio State University ULAR and National Institutes of Health (NIH) animal facilities in compliance with Institutional Animal Care and Use Committee (IACUC) standards. All animal experiments were conducted in accordance with the ethical guidelines under the protocols OSU IACUC 2022A00000061 and NIAID LISB-13E.

Cell cultures were maintained in sterile complete Roswell Park Memorial Institute (RPMI) media. The RPMI 1640 medium was supplemented with 10% heat-inactivated fetal bovine serum, 50 U/ml penicillin, 50 μM streptomycin, 1 mM sodium pyruvate, 2 mM L-glutamine, 0.1 mM non-essential amino acids, 50 μM 2-mercaptoethanol, and 10 mM HEPES (Thermo Fisher Scientific). For magnetic separations, cells were maintained in filtered and degassed magnetic-activated cell sorting (MACS) buffer. The MACS buffer consisted of PBS (Lonza) supplemented with 0.5% BSA (Sigma-Aldrich) and 2 mM EDTA (Sigma-Aldrich). For splenic DC isolation, Liberase TM (Blendzyme) and DNase I were purchased from Roche. 50 mg Liberase was dissolved in complete RPMI, resulting in a final concentration of 5 mg/mL. The solution was kept on ice, stirred gently every 5 min for 30 min, and aliquoted into 1 mL portions. 10 mg/mL DNAse stock solution was prepared with 0.15 M NaCl and divided into 500 μL aliquots. Both enzymes were stored at –20 °C.

Samples were stained for flow cytometry using fluorescence-activated cell sorting (FACS) buffer (either PBS or Hanks' balanced salt solution (HBSS) supplemented with 2% fetal bovine serum,

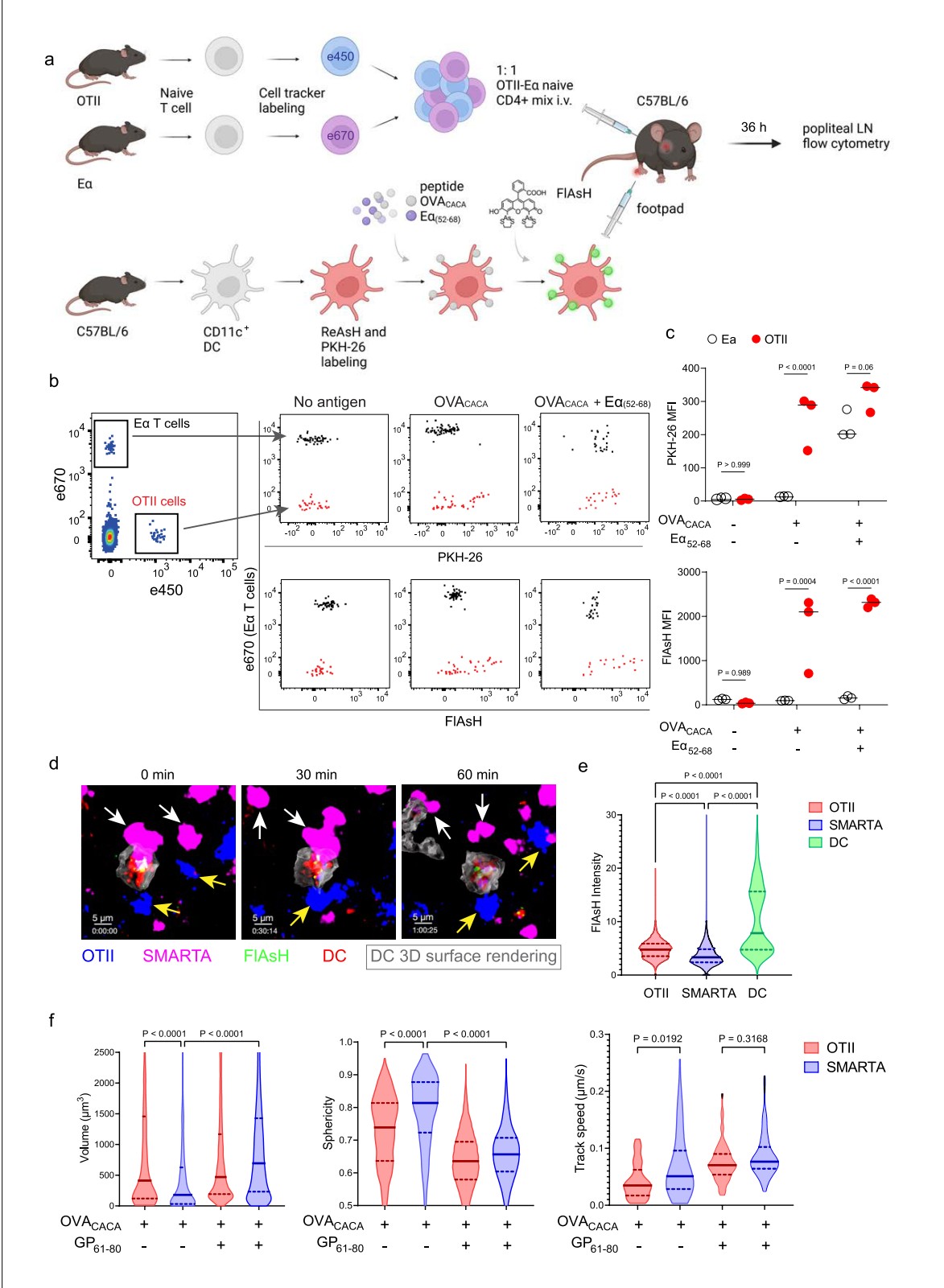

**Figure 5.** FlAsH-pMHCII is incorporated by cognate T cells in an antigen-specific manner. (**a–c**) Dendritic cells (DCs) were labeled with ReAsH and PKH-26, single/ double-pulsed with 5 μM OVA$_{CACA}$, Eα$_{(52-68)}$, or left unpulsed. They were labeled with Fluorescein-AsH (FlAsH) and adoptively transferred into wild-type (WT) mice (0.5–1 × 10$^6$ cells) via footpad. Naïve e450-labeled OTII and e670-labeled Eα-specific T cells were mixed 1:1 (0.25–0.5 × 10$^6$ / T cell type) and injected i.v. Popliteal lymph node was analyzed by flow cytometry at 36–42 hr post-transfer. (**a**) Experimental scheme. (**b**) Summary plots.

*Figure 5 continued on next page*

*Figure 5 continued*

(**c**) PKH-26 and FlAsH MFI in OTII and Eα-specific T cells. Data are representative of two independent experiments with n=3 mice per group. p-values were calculated using two-way ANOVA with Sidak's multiple comparisons test. (**d-f**) DCs were labeled with ReAsH and single/double-pulsed with 5 μM OVA$_{CACA}$, LCMV GP$_{61-80}$, labeled with FlAsH and adoptively transferred into WT mice (1–2×10$^6$ cells) via footpad. Naïve e450-labeled OTII and e670-labeled SMARTA T cells were mixed 1:1 (0.5–1×10$^6$ / T cell type) and injected i.v. Live popliteal lymph node sections were imaged at 18 hr. (**d**) Time series representative of DC-T cell interactions. (**e**) Average Flash intensity of the adoptively transferred cells. (**f**) T cell blasting (left and middle panels) and motility (right) following adoptive transfer with DCs pulsed with the indicated peptides. Data are representative of three independent experiments with n=2 mice per group. p-values were calculated using one-way ANOVA with Tukey's multiple comparisons test.

1% HEPES, and 10 mM sodium azide (Sigma-Aldrich)). For confocal microscopy, PBS supplemented with 1% BSA was used. The antibodies used for flow cytometry are as follows: anti-CD4-efluor450 (clone GK1.5), anti-B220-PE (clone RA3-6B2), anti-CD11c-AF700 (clone N418), anti-CD45.1-PE.Cy7 (clone A20, Thermo Fisher Scientific), anti-CD45.2-AF700 (clone 104), anti-CD44-BUV395 (clone IM7, BD Biosciences), anti-I-A/I-E-BV605 (clone M5/114.15.2), anti-I-A/I-E-BV785 (clone M5/114.15.2), streptavidin-BV650, streptavidin-AF647 and purified anti-CD16/32 (clone 93). The antibodies were purchased from BioLegend unless stated otherwise. All peptides used in the study (*Supplementary file 1*) were obtained from the NIH Research Technologies Branch, NIAID Peptide Core Facility.

For AsH probe labeling, peptides were pre-treated with fresh Tris Carboxy Ethyl Phosphene (TCEP, Sigma-Aldrich) to protect tetracysteine residues from oxidation. 10 mM ReAsH stock solution was prepared by adding 45 μL sterile cell-culture grade DMSO (Sigma-Aldrich) to 250 μg solid ReAsH (Cayman Chemical) and 10 mM FlAsH solution was prepared by adding 150 uL DMSO to 1 mg FlAsH powder (Cayman Chemical). British Anti-Lewisite (BAL, Alfa Aesar) and HBSS with Ca$^{++}$ and Mg$^{++}$ (Thermo Fisher Scientific) was used to wash excess FlAsH off the cells. BAL is resuspended in sterile degassed water to obtain 8 mM stock solution. FlAsH, ReAsH, and BAL are highly sensitive to oxidation, therefore, their activity may decline in subsequent uses. This may be averted, at least in part, by quickly purging stock solution containers with 4 M Argon gas (Sigma-Aldrich, 501247), placing container in a 50 mL tube, purging again with Argon before storing FlAsH and ReAsH at –20 °C, BAL at 4 °C. We do not recommend storage longer than one month or repeated use of the same reagent stock.

## DC isolation, peptide loading, and -AsH probe labeling

Mature splenic DCs were isolated as previously described (*Akkaya et al., 2021*) and resuspended in complete RPMI. Peptides were added into 20 mM TCEP solution at twice the desired concentration for loading DCs. The solution was vortexed thoroughly and incubated at 37 °C for 30 min. Meanwhile, ReAsH stock solution was diluted to 100 uM in complete RPMI and added on DC suspension to obtain 1 uM final ReAsH concentration. Cells were incubated at 37 °C for 30 min and washed twice with complete RPMI. ReAsH-labeled DCs were mixed with an equal volume of peptide-TCEP solution, incubated at 37 °C for 30 min, washed twice, and resuspended in complete RPMI to proceed with FlAsH labeling. Flash was added to cell suspension to achieve 1 uM final concentration, incubated at 37 °C for 30 min and washed twice with complete RPMI. To remove excess FlAsH and reduce non-specific

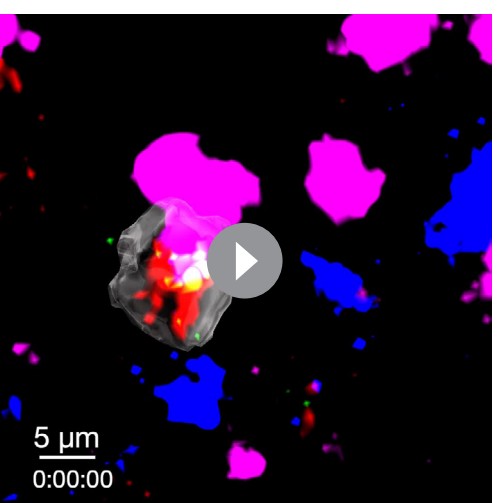

**Video 2.** ReAsH-labeled dendritic cells (DCs) were double-pulsed with OVA$_{CACA}$ and LCMV GP$_{61-80}$, treated with Fluorescein-AsH (FlAsH) (green) and adoptively transferred via footpad. Naïve e450-labeled OTII and e670-labeled SMARTA T cells were mixed 1:1 and injected i.v. via retroorbital sinus. Popliteal lymph nodes of the recipient mice were surgically removed 18 hr after transfer, sectioned, and recovered at 37 °C until the resumption of cell migration in the tissue. Sections were then imaged at 37 °C by confocal microscopy for 4 hr. Video shows that both OTII (blue) and SMARTA T cells (magenta) have sustained contacts with the DC (Red). Gray: 3D surface rendering for the DCs. Data are representative of three independent experiments.
https://elifesciences.org/articles/91809/figures#video2

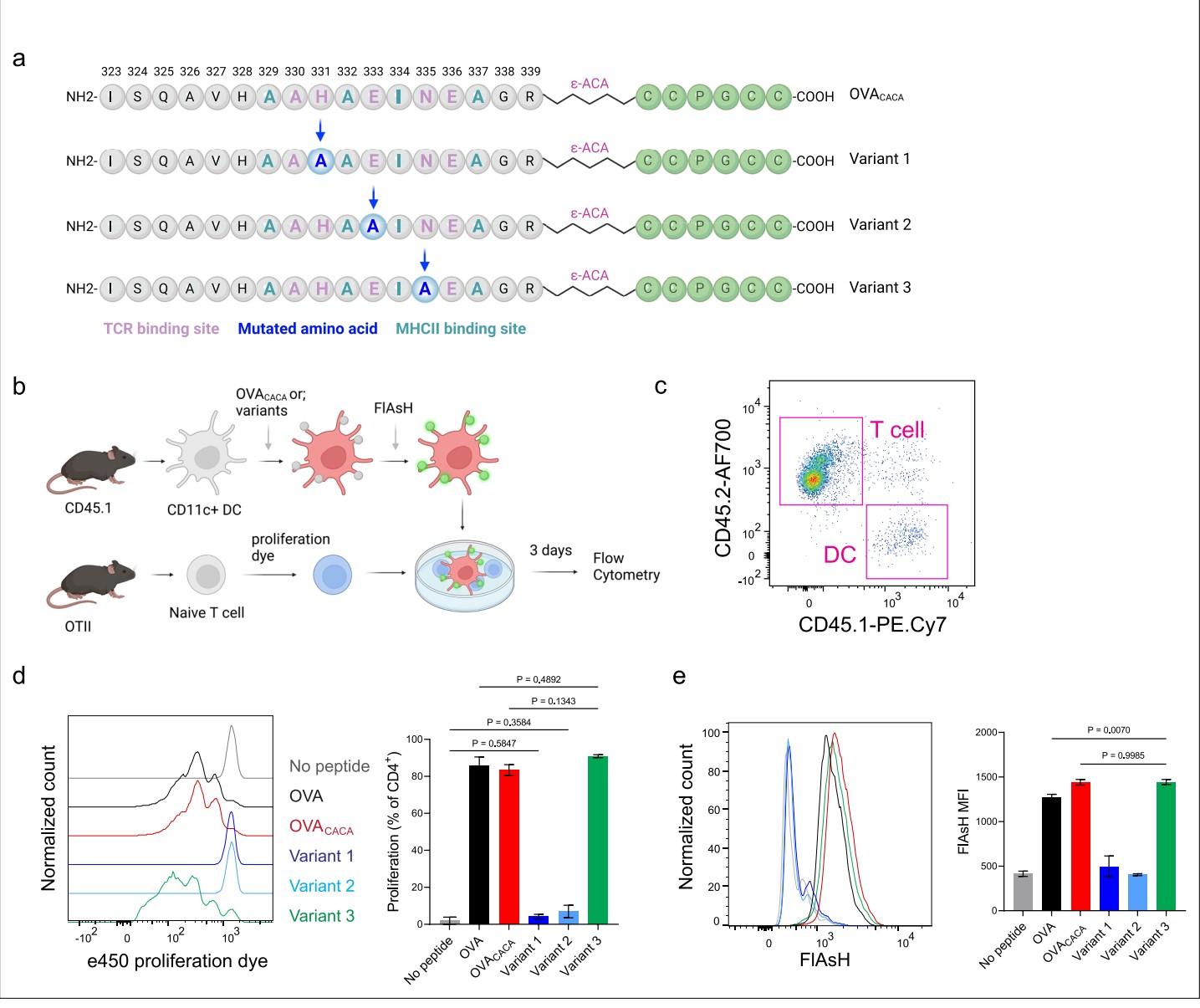

**Figure 6.** Nuanced TCR-pMHCII interactions can be discerned by Fluorescein-AsH (FlAsH) labeling of peptides. (**a–b**) Graphical depiction of the alanine-mutants for OVA$_{CACA}$ TCR binding site and experimental approach. $0.5 \times 10^5$ CD45.1$^+$ DCs were pulsed with the indicated peptides, labeled with FlAsH, and cocultured with $2.5 \times 10^5$ e450-labeled OTII cells for 3 days. (**c**) Gating strategy for T cells and dendritic cell (DCs). (**d**) Representative flow cytometry plots for OTII T cell proliferation with a partial agonist (top) and two antagonist peptides (bottom). (**e**) FlAsH signal intensity of DCs and OTII cells with partial agonist (top) and antagonist peptides (bottom) adoptive transfer with the indicated peptides followed by FlAsH treatment. Data are representative of two independent experiments performed with n=3 biological replicates.

binding, cells were resuspended in 240 uM BAL diluted in HBSS with Ca$^{++}$ and Mg$^{++}$, washed twice, then washed a third time with PBS, and resuspended in PBS.

## DC-T cell cocultures

Freshly isolated DCs were pulsed with peptide in complete RPMI and were incubated at 37 °C for 30 min followed by three washes with complete RPMI to remove unbound peptide. Pulsed DCs were then treated with -AsH as described above and then seeded into flat-bottom 96-well plates (Corning) at a density of $5 \times 10^4$–$1 \times 10^5$ cells per well. T cells and iTreg cells were labeled with e450 or e670 (eBioscience) according to the manufacturer's protocol. Cells were cocultured at 37 °C for indicated periods, at a 1:10 ratio of DCs to T cells for proliferation assays and 1:1 for trogocytosis assays.

## iTreg differentiation

For iTreg differentiation, Tnaive cells were isolated, and 24-well sterile tissue culture plates (Corning) were coated with anti-CD3ε (BioLegend) and anti-CD28 (BioLegend) as described (*Akkaya et al., 2017*). Naïve CD4+ T cells were resuspended in complete RPMI media supplemented with 100 IU ml−1 recombinant human IL-2 (Peprotech) and 5 ng ml−1 recombinant human TGF-β (Peprotech). $3 \times 10^5$ cells were added to the wells at a volume of 1 ml per well. Cells were cultured at 37 °C, 5% CO2 for 3–4 days.

## Trogocytosis assay

Trogocytosis assays were performed as described in *Akkaya et al., 2019*. Briefly, freshly isolated splenic DCs were labeled with 4 µM PKH26 (Sigma-Aldrich) for labeling of the cell membrane as described in Puaux et al., DCs were then pulsed with 10 µM peptide, treated with -AsH, and were cocultured with T cells for up to 18 hr. Cell conjugates were dissociated by washing cells with MACS buffer with 2 mM EDTA; cell suspensions were prepared for flow cytometry.

## Flow cytometry

Cells were stained in FACS buffer containing fluorochrome-conjugated antibodies for 30 min at 4 °C, protected from light. Data acquisition was performed with BD LSRFortessa and BD LSR II cytometers (BD Biosciences). Data analysis was performed with FlowJo v10.8.1.

## Adoptive transfer

Mice were anesthetized in an induction chamber with 1.5% isoflurane USP (Baxter). Pulsed DCs were adoptively transferred through the right footpad in 50 uL sterile PBS. T cells labeled with e450 or e670 (Thermo Fisher Scientific) were injected intravenously in 100 uL sterile PBS through the retro-orbital sinus.

## Confocal microscopy

35 mm dishes with a 14 mm glass coverslip bottom (MatTek) were pre-treated with 10 ug/ml fibronectin (Sigma-Aldrich) in PBS for 1 hr at room temperature and washed twice. Splenic DCs were added onto the glass coverslip in complete RPMI and pulsed with 5–10 µM peptide at 37 °C for 1 hr. Excess peptide was washed off with complete RPMI before the addition of CD4 T cells. Co-cultures were imaged for 3–18 hr using a Leica SP-8 inverted microscope (Leica Microsystems) equipped with a full range of visible lasers, two hybrid detectors, three photomultiplier detectors, and a motorized stage. 63x objective (Leica Microsystems) was used and microscope configuration was set up for 3D analysis (x, y, z) of the cellular layer. The following lasers were used: diode laser for 405 nm excitation; argon laser for 488 and 514 nm excitation; diode-pumped solid-state laser for 561 nm; and HeNe laser for 594 and 633 nm excitation. All lasers were tuned to minimal power (between 0.3% and 2%) to prevent photobleaching. z stacking of images of 10–12 µm were collected. Mosaic images of large cell culture areas (1 mm²) were generated by acquiring multiple z stacks using the Tile scan mode and were assembled into tiled images using LAS X v4.0 (Leica Microsystems).

## Intravital two-photon laser-scanning microscopy of mouse popliteal lymph nodes

Mice were anesthetized in 1.5% isoflurane USP (Baxter) through a rodent nose cone followed by surgery to expose the right popliteal lymph node. Warm PBS was added to maintain lymph node moisture. Mouse body temperature was maintained with an infrared blanket (Braintree Scientific) throughout the imaging process after which they were euthanized through cervical dislocation under anesthesia. Mice were imaged on a glass stage with a two-photon laser-scanning microscopy setup with a Leica SP8 inverted confocal microscope with dual multi-photon lasers, Mai Tai and InSight DS (Spectra-Physics), L 25.0 water-immersion objective, 0.95 NA (Leica Microsystems), and a 37 °C incubation chamber (NIH, Division of Scientific Equipment and Instrumentation Services). The Mai Tai laser was tuned to 890 nm to excite e450 and FlAsH; the InSight DS laser was tuned to 1150 nm to excite DsRed/ PKH-26 /ReAsH and e670. For time-lapse imaging, a z stack consisting of 10–12 single planes

(5 μm each over a total tissue depth of 50–60 μm) was acquired at 15 s intervals for a total period of 1–4 hr.

## Two-photon laser-scanning microscopy of live lymph node sections

Live lymph node sections were imaged ex vivo by two-photon microscopy. To maintain tissue architecture, popliteal lymph nodes were immediately transferred to PBS with 1% BSA and kept on ice immediately following harvest. Residual fat and connective tissue were removed using a Leica MZ6 StereoZoom microscope (Leica Microsystems). Lymph nodes were suspended at 38 °C agarose in DMEM and kept over ice. Following complete agarose solidification, complete RPMI was added and blocks were cut containing one lymph node each. Lymph node blocks were cut into 200 μm sections with a Leica VT1000 S vibrating blade microtome (Leica Microsystems) at speed 5 in ice-cold PBS and frozen at –80 °C. Prior to imaging, a complete RPMI medium was added to tissue sections and incubated at 37 °C for 2 hr. Sections were held down with tissue anchors (Warner Instruments) in 35 mm dishes with a 14 mm glass coverslip bottom (MatTek) and were imaged with a Leica SP8 inverted five-channel confocal microscope equipped with an environmental chamber (NIH, Division of Scientific Equipment and Instrumentation Services) and a motorized stage. The microscope configuration was set up for four-dimensional analysis (x, y, z, t) of cell segregation and migration within tissue sections. A diode laser for 405 nm, an argon laser for 488 and 514 nm, a diode-pumped solid-state laser for 561 nm, and a HeNe laser for 594 and 633 nm excitation wavelengths were tuned to minimal power (between 0.3% and 2%). z-stack of images of 10–25 μm were collected. Mosaic images of whole lymph nodes were generated by acquiring multiple z stacks using a motorized stage to cover the whole lymph node area and assembled into a tiled image with the LAS X software. For time-lapse analysis of cell migration, tiled z-stacks were collected over time (1–4 hr).

## Image analysis

Analysis of confocal and intravital two-photon images was performed using Imaris v9.2.1 and v10.0.0 (Bitplane) as previously described (*Akkaya et al., 2021*). In brief, e450 and e670 signals were utilized for reconstructing the 3D structure of T cells and iTreg cells as surface objects. To reconstruct the 3D structure of DCs, overlapping DsRed, PKH-26, and ReAsH signals were employed. Imaris v10.0.0's built-in algorithms were used to calculate 3D volumes, track velocities, surface-to-surface colocalization parameters, and FlAsH (green) intensities of cells.

## Statistical analysis

Statistical analysis was conducted on Prism v9.5.1 (GraphPad). Statistical tests and relevant p-values for each figure are indicated in the figure or figure legend.

## Acknowledgements

This study was funded by was by the National Institutes of Allergy and Infectious Diseases Intramural Research Grant ZIAAI000224, Extramural Research Grant DP2AI154451, and research funds from the Ohio State University College of Medicine. We thank Arpad Somogyi and Aktham Mestareehi for their insightful comments and suggestions.

## Additional information

### Funding

| Funder | Grant reference number | Author |
| --- | --- | --- |
| National Institute of Allergy and Infectious Diseases | DP2AI154451 | Billur Akkaya |
| National Institute of Allergy and Infectious Diseases | ZIAAI000224 | Ethan Shevach |

| Funder | Grant reference number | Author |
|---|---|---|

The funders had no role in study design, data collection and interpretation, or the decision to submit the work for publication.

### Author contributions

Munir Akkaya, Conceptualization, Supervision, Investigation, Methodology, Writing – review and editing; Jafar Al Souz, Formal analysis, Investigation, Methodology, Writing – review and editing; Daniel Williams, Formal analysis, Investigation, Methodology; Rahul Kamdar, Olena Kamenyeva, Investigation, Methodology; Juraj Kabat, Formal analysis; Ethan Shevach, Supervision, Writing – review and editing; Billur Akkaya, Conceptualization, Resources, Formal analysis, Supervision, Funding acquisition, Investigation, Methodology, Writing – original draft, Project administration, Writing – review and editing

### Author ORCIDs

Olena Kamenyeva http://orcid.org/0000-0002-6541-5616
Billur Akkaya http://orcid.org/0000-0002-6808-3776

### Ethics

This study was performed in strict accordance with the recommendations in the Guide for the Care and Use of Laboratory Animals of the National Institutes of Health and in compliance with Animal Care and Use Committee standards.

Reviewer #1 (Public Review): https://doi.org/10.7554/eLife.91809.3.sa1
Reviewer #2 (Public Review): https://doi.org/10.7554/eLife.91809.3.sa2
Author Response https://doi.org/10.7554/eLife.91809.3.sa3

## Additional files

### Supplementary files

• Supplementary file 1. The sequences and molecular weights of the peptides used in the study.
• MDAR checklist

### Data availability

The flow cytometry and imaging data that support the findings of this study are available in Dryad at https://doi.org/10.5061/dryad.v15dv4239.

The following dataset was generated:

| Author(s) | Year | Dataset title | Dataset URL | Database and Identifier |
|---|---|---|---|---|
| Akkaya M, Souz JAl, Williams D, Kamdar R, Kamenyeva O, Kabat J, Shevach EM, Akkaya B | 2023 | Illuminating T cell-dendritic cell interactions in vivo by FlAsHing antigens | https://doi.org/ 10.5061/dryad. v15dv4239 | Dryad Digital Repository, 10.5061/dryad.v15dv4239 |

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
