## [Editor Report · eLife assessment]

This is an **important** study that develops a method to fluorescently label peptide MHC complexes on live dendritic cells to enable detection of antigen specific T cells in polyclonal populations. **Solid** evidence that this can be used to effectively identify antigen specific T cells in vitro and in vivo is provided for one model antigen systems (Ova-OTII). The approach has exciting potential as prior single step methods with directly conjugated single peptides have generally failed due to high background. Thus, this approach potentially moves the state of the art forward, but further work is needed to realise and determine the limits and ultimate utility of the approach.

---

## [Referee Report · Reviewer #1 (Public Review)]

Summary:

The authors develop a method to fluorescently tagged peptides loaded onto dendritic cells using a three step method. These include a pre blocking step to block endogenous cysteine motifs on the DC surface, loading a tetracystein motif modified peptide on surface MHC and a labelling step done on the surface of live DC using a dye with high affinity for the added motif. The results are convincing in demonstrating in vitro and in vivo T cell activation and efficient label transfer to specific T cells in vivo. The label transfer technique will be useful to identify T cell that have recognised a DC presenting a specific peptide antigen to allow the isolation of the T cell and cloning of its TCR subunits, for example. It may also be useful as a general assay for in vitro or in vivo T-DC communication that can allow detection of genetic or chemical modulators.

Strengths:

The study include both in vitro and in vivo analysis including flow cytometry and two photon laser scanning microscopy. The results are convincing and the level of T cell labelling with the fluorescent pMHC is surprisingly robust and suggests that the approach is potentially revealing something about fundamental mechanisms beyond the state of the art. They also provide practical information about the challenges of the method and discuss limitations.

Weaknesses:

The method is demonstrated only at high pMHC density and it would need to be re-optimised to determine if it can be used at lower densities that may often be encountered physiologically.

---

## [Referee Report · Reviewer #2 (Public Review)]

Summary:

The authors have developed novel Ovalbumin model (OTII) peptide that can be labeled with a site-specific FlAsH dye to track agonist peptides both in vitro and in vivo. The utility of this tool could allow better tracking of activated polyclonal T cells particularly in novel systems. The authors have provided solid evidence that peptides are functional, capable of activating OTII T cells, and these peptides can undergo trogocytosis by cognate T cells only.

Strengths:

-An extensive array of in vitro and in vivo studies are used to assess peptide functionality.

-Nice use of cutting edge intravital imaging,

-internal controls such as multiple non-cogate T cells were used to improve robustness of the results

-One of the strengths is the direct labeling of the peptide, and the potential utility in other systems.

Weaknesses:

-Peptide labeling specificity and efficiency is not clear. High levels of background labeling. While it was sufficient for demonstrating the system works, it may pose problems depending on the peptide sequence, and/or use at lower dose.

-Only one peptide system was tested, namely OVA323-339 region.

-Limited novel biological findings. This study mostly describes a new tool that may have exciting potential.

---

## [Author Response]

The following is the authors’ response to the original reviews.

**Public Reviews:**

**Reviewer #1 (Public Review):**
Summary:The authors develop a method to fluorescently tag peptides loaded onto dendritic cells using a two-step method with a tetracystein motif modified peptide and labelling step done on the surface of live DC using a dye with high affinity for the added motif. The results are convincing in demonstrating in vitro and in vivo T cell activation and efficient label transfer to specific T cells in vivo. The label transfer technique will be useful to identify T cells that have recognised a DC presenting a specific peptide antigen to allow the isolation of the T cell and cloning of its TCR subunits, for example. It may also be useful as a general assay for in vitro or in vivo T-DC communication that can allow the detection of genetic or chemical modulators.Strengths:The study includes both in vitro and in vivo analysis including flow cytometry and two-photon laser scanning microscopy. The results are convincing and the level of T cell labelling with the fluorescent pMHC is surprisingly robust and suggests that the approach is potentially revealing something about fundamental mechanisms beyond the state of the art.Weaknesses:The method is demonstrated only at high pMHC density and it is not clear if it can operate at at lower peptide doses where T cells normally operate. However, this doesn't limit the utility of the method for applications where the peptide of interest is known. It's not clear to me how it could be used to de-orphan known TCR and this should be explained if they want to claim this as an application. Previous methods based on biotin-streptavidin and phycoerythrin had single pMHC sensitivity, but there were limitations to the PE-based probe so the use of organic dyes could offer advantages.

We thank the reviewer for the valuable comments and suggestions. Indeed, we have shown and optimized this labeling technique for a commonly used peptide at rather high doses to provide a proof of principle for the possible use of tetracysteine tagged peptides for in vitro and in vivo studies. However, we completely agree that the studies that require different peptides and/or lower pMHC concentrations may require preliminary experiments if the use of biarsenical probes is attempted. We think it can help investigate the functional and biological properties of the peptides for TCRs deorphaned by techniques. Tetracysteine tagging of such peptides would provide a readily available antigen-specific reagent for the downstream assays and validation. Other possible uses for modified immunogenic peptides could be visualizing the dynamics of neoantigen vaccines or peptide delivery methods in vivo. For these additional uses, we recommend further optimization based on the needs of the prospective assay.

**Reviewer #2 (Public Review):**
Summary:The authors here develop a novel Ovalbumin model peptide that can be labeled with a site-specific FlAsH dye to track agonist peptides both in vitro and in vivo. The utility of this tool could allow better tracking of activated polyclonal T cells particularly in novel systems. The authors have provided solid evidence that peptides are functional, capable of activating OTII T cells, and that these peptides can undergo trogocytosis by cognate T cells only.Strengths:-An array of in vitro and in vivo studies are used to assess peptide functionality.-Nice use of cutting-edge intravital imaging.-Internal controls such as non-cogate T cells to improve the robustness of the results (such as Fig 5A-D).-One of the strengths is the direct labeling of the peptide and the potential utility in other systems.Weaknesses:1. What is the background signal from FlAsH? The baselines for Figure 1 flow plots are all quite different. Hard to follow. What does the background signal look like without FLASH (how much fluorescence shift is unlabeled cells to No antigen+FLASH?). How much of the FlAsH in cells is actually conjugated to the peptide? In Figure 2E, it doesn't look like it's very specific to pMHC complexes. Maybe you could double-stain with Ab for MHCII. Figure 4e suggests there is no background without MHCII but I'm not fully convinced. Potentially some MassSpec for FLASH-containing peptides.

We thank the reviewer for pointing out a possible area of confusion. In fact, we have done extensive characterization of the background and found that it has varied with the batch of FlAsH, TCEP, cytometer and also due to the oxidation prone nature of the reagents. Because Figure 1 subfigures have been derived from different experiments, a combination of the factors above have likely contributed to the inconsistent background. To display the background more objectively, we have now added the No antigen+Flash background to the revised Fig 1.

It is also worthwhile noting that nonspecific Flash incorporation can be toxic at increasing doses, and live cells that display high backgrounds may undergo early apoptotic changes in vitro. However, when these cells are adoptively transferred and tracked in vivo, the compromised cells with high background possibly undergo apoptosis and get cleared by macrophages in the lymph node. The lack of clearance in vitro further contributes to different backgrounds between in vitro and in vivo, which we think is also a possible cause for the inconsistent backgrounds throughout the manuscript. Altogether, comparison of absolute signal intensities from different experiments would be misleading and the relative differences within each experiment should be relied upon. We have added further discussion about this issue.

1. On the flip side, how much of the variant peptides are getting conjugated in cells? I'd like to see some quantification (HPLC or MassSpec). If it's ~10% of peptides that get labeled, this could explain the low shifts in fluorescence and the similar T cell activation to native peptides if FlasH has any deleterious effects on TCR recognition. But if it's a high rate of labeling, then it adds confidence to this system.

We agree that mass spectrometry or, more specifically tandem MS/MS, would be an excellent addition to support our claim about peptide labeling by FlAsH being reliable and non-disruptive. Therefore, we have recently undertaken a tandem MS/MS quantitation project with our collaborators. However, this would require significant time to determine the internal standard based calibration curves and to run both analytical and biological replicates. Hence, we have decided pursuing this as a follow up study and added further discussion on quantification of the FlAsH-peptide conjugates by tandem MS/MS.

1. Conceptually, what is the value of labeling peptides after loading with DCs? Why not preconjugate peptides with dye, before loading, so you have a cleaner, potentially higher fluorescence signal? If there is a potential utility, I do not see it being well exploited in this paper. There are some hints in the discussion of additional use cases, but it was not clear exactly how they would work. One mention was that the dye could be added in real-time in vivo to label complexes, but I believe this was not done here. Is that feasible to show?

We have already addressed preconjugation as a possible avenue for labeling peptides. In our hands, preconjugation resulted in low FlAsH intensity overall in both the control and tetracysteine labeled peptides (Author response image 1). While we don’t have a satisfactory answer as to why the signal was blunted due to preconjugation, it could be that the tetracysteine tagged peptides attract biarsenical compounds better intracellularly. It may be due to the redox potential of the intracellular environment that limits disulfide bond formation. (PMID: 18159092)

**Author response image 1. sa3fig1:** Preconjugation yields poor FlAsH signal. Splenic DCs were pulsed with peptide then treated with FlAsH or incubated with peptide-FlAsH preconjugates. Overlaid histograms show the FlAsH intensities on DCs following the two-step labeling (left) and preconjugation (right). Data are representative of two independent experiments, each performed with three biological replicates.

1. Figure 5D-F the imaging data isn't fully convincing. For example, in 5F and 2G, the speeds for T cells with no Ag should be much higher (10-15micron/min or 0.16-0.25micron/sec). The fact that yours are much lower speeds suggests technical or biological issues, that might need to be acknowledged or use other readouts like the flow cytometry.

We thank the reviewer for drawing attention to this technical point. We would like to point out that the imaging data in fig 5 d-f was obtained from agarose embedded live lymph node sections. Briefly, the lymph nodes were removed, suspended in 2% low melting temp agarose in DMEM and cut into 200 µm sections with a vibrating microtome. Prior to imaging, tissue sections were incubated in complete RPMI medium at 37 °C for 2 h to resume cell mobility. Thus, we think the cells resuming their typical speeds ex vivo may account for slightly reduced T cell speeds overall, for both control and antigen-specific T cells (PMID: 32427565, PMID: 25083865). We have added text to prevent the ambiguity about the technique for dynamic imaging. The speeds in Figure 2g come from live imaging of DC-T cell cocultures, in which the basal cell movement could be hampered by the cell density. Additionally, glass bottom dishes have been coated with Fibronectin to facilitate DC adhesion, which may be responsible for the lower average speeds of the T cells in vitro.

**Reviewer #1 (Recommendations For The Authors):**
Does the reaction of ReAsH with reactive sites on the surface of DC alter them functionally? Functions have been attributed to redox chemistry at the cell surface- could this alter this chemistry?

We thank the reviewer for the insight. It is possible that the nonspecific binding of biarsenical compounds to cysteine residues, which we refer to as background throughout the manuscript, contribute to some alterations. One possible way biarsenicals affect the redox events in DCs can be via reducing glutathione levels (PMID: 32802886). Glutathione depletion is known to impair DC maturation and antigen presentation (PMID: 20733204). To avoid toxicity, we have carried out a stringent titration to optimize ReAsH and FlAsH concentrations for labeling and conducted experiments using doses that did not cause overt toxicity or altered DC function.

Have the authors compared this to a straightforward approach where the peptide is just labelled with a similar dye and incubated with the cell to load pMHC using the MHC knockout to assess specificity? Why is this that involves exposing the DC to a high concentration of TCEP, better than just labelling the peptide? The Davis lab also arrived at a two-step method with biotinylated peptide and streptavidin-PE, but I still wonder if this was really necessary as the sensitivity will always come down to the ability to wash out the reagents that are not associated with the MHC.

We agree with the reviewer that small undisruptive fluorochrome labeled peptide alternatives would greatly improve the workflow and signal to noise ratio. In fact, we have been actively searching for such alternatives since we have started working on the tetracysteine containing peptides. So far, we have tried commercially available FITC and TAMRA conjugated OVA323-339 for loading the DCs, however failed to elicit any discernible signal. We also have an ongoing study where we have been producing and testing various in-house modified OVA323-339 that contain fluorogenic properties. Unfortunately, at this moment, the ones that provided us with a crisp, bright signal for loading revealed that they have also incorporated to DC membrane in a nonspecific fashion and have been taken up by non-cognate T cells from double antigen-loaded DCs. We are actively pursuing this area of investigation and developing better optimized peptides with low/non-significant membrane incorporation.

Lastly, we would like to point out that tetracysteine tags are visible by transmission electron microscopy without FlAsH treatment. Thus, this application could add a new dimension for addressing questions about the antigen/pMHCII loading compartments in future studies. We have now added more in-depth discussion about the setbacks and advantages of using tetracysteine labeled peptides in immune system studies.

The peptide dosing at 5 µM is high compared to the likely sensitivity of the T cells. It would be helpful to titrate the system down to the EC50 for the peptide, which may be nM, and determine if the specific fluorescence signal can still be detected in the optimal conditions. This will not likely be useful in vivo, but it will be helpful to see if the labelling procedure would impact T cell responses when antigen is limited, which will be more of a test. At 5 µM it's likely the system is at a plateau and even a 10-fold reduction in potency might not impact the T cell response, but it would shift the EC50.

We thank the reviewer for the comment and suggestion. We agree that it is possible to miss minimally disruptive effects at 5 µM and titrating the native peptide vs. modified peptide down to the nM doses would provide us a clearer view. This can certainly be addressed in future studies and also with other peptides with different affinity profiles. A reason why we have chosen a relatively high dose for this study was that lowering the peptide dose had costed us the specific FlAsH signal, thus we have proceeded with the lowest possible peptide concentration.

In Fig 3b the level of background in the dsRed channel is very high after DC transfer. What cells is this associated with and does this appear be to debris? Also, I wonder where the ReAsH signal is in the experiments in general. I believe this is a red dye and it would likely be quite bright given the reduction of the FlAsH signal. Will this signal overlap with signals like dsRed and PHK-26 if the DC is also treated with this to reduce the FlAsH background?

We have already shown that ReAsH signal with DsRed can be used for cell-tracking purposes as they don’t get transferred to other cells during antigen specific interactions (Author response image 2). In fact, combining their exceptionally bright fluorescence provided us a robust signal to track the adoptively transferred DCs in the recipient mice. On the other hand, the lipophilic membrane dye PKH-26 gets transferred by trogocytosis while the remaining signal contributes to the red fluorescence for tracking DCs. Therefore, the signal that we show to be transferred from DCs to T cells only come from the lipophilic dye. To address this, we have added a sentence to elaborate on this in the results section. Regarding the reviewer’s comment on DsRed background in Figure 3b., we agree that the cells outside the gate in recipient mice seems slightly higher that of the control mice. It may suggest that the macrophages clearing up debris from apoptotic/dying DCs might contribute to the background elicited from the recipient lymph node. Nevertheless, it does not contribute to any DsRed/ReAsH signal in the antigen-specific T cells.

**Author response image 2. sa3fig2:** ReAsH and DsRed are not picked up by T cells during immune synapse. DsRed+ DCs were labeled with ReAsH, pulsed with 5 μM OVACACA, labeled with FlAsH and adoptively transferred into CD45.1 congenic mice mice (1-2 × 106 cells) via footpad. Naïve e450-labeled OTII and e670-labeled polyclonal CD4+ T cells were mixed 1:1 (0.25-0.5 × 106/ T cell type) and injected i.v. Popliteal lymph nodes were removed at 42 h post-transfer and analyzed by flow cytometry. Overlaid histograms show the ReAsh/DsRed, MHCII and FlAsH intensities of the T cells. Data are representative of two independent experiments with n=2 mice per group.

In Fig 5b there is a missing condition. If they look at Ea-specific T cells for DC with without the Ova peptide do they see no transfer of PKH-26 to the OTII T cells? Also, the FMI of the FlAsH signal transferred to the T cells seems very high compared to other experiments. Can the author estimate the number of peptides transferred (this should be possible) and would each T cell need to be collecting antigens from multiple DC? Could the debris from dead DC also contribute to this if picked up by other DC or even directly by the T cells? Maybe this could be tested by transferring DC that are killed (perhaps by sonication) prior to inoculation?

To address the reviewer’s question on the PKH-26 acquisition by T cells, Ea-T cells pick up PKH-26 from Ea+OVA double pulsed DCs, but not from the unpulsed or single OVA pulsed DCs. OTII T cells acquire PKH-26 from OVA-pulsed DCs, whereas Ea T cells don’t (as expected) and serve as an internal negative control for that condition. Regarding the reviewer’s comment on the high FlAsH signal intensity of T cells in Figure 5b, a plausible explanation can be that the T cells accumulate pMHCII through serial engagements with APCs. In fact, a comparison of the T cell FlAsH intensities 18 h and 36-48 h post-transfer demonstrate an increase (Author response image 3) and thus hints at a cumulative signal. As DCs are known to be short-lived after adoptive transfer, the debris of dying DCs along with its peptide content may indeed be passed onto macrophages, neighboring DCs and eventually back to T cells again (or for the first time, depending on the T:DC ratio that may not allow all T cells to contact with the transferred DCs within the limited time frame). We agree that the number and the quality of such contacts can be gauged using fluorescent peptides. However, we think peptides chemically conjugated to fluorochromes with optimized signal to noise profiles and with less oxidation prone nature would be more suitable for quantification purposes.

**Author response image 3. sa3fig3:** FlAsH signal acquisition by antigen specific T cells becomes more prominent at 36-48 h post-transfer. DsRed+ splenic DCs were double-pulsed with 5 μM OVACACA and 5 μM OVA-biotin and adoptively transferred into CD45.1 recipients (2 × 106 cells) via footpad. Naïve e450-labeled OTII (1 × 106 cells) and e670-labeled polyclonal T cells (1 × 106 cells) were injected i.v. Popliteal lymph nodes were analyzed by flow cytometry at 18 h or 48 h post-transfer. Overlaid histograms show the T cell levels of OVACACA (FlAsH). Data are representative of three independent experiments with n=3 mice per time point

**Reviewer #2 (Recommendations For The Authors):**
As mentioned in weaknesses 1 & 2, more validation of how much of the FlAsH fluorescence is on agonist peptides and how much is non-specific would improve the interpretation of the data. Another option would be to preconjugate peptides but that might be a significant effort to repeat the work.

We agree that mass spectrometry would be the gold standard technique to measure the percentage of tetracysteine tagged peptide is conjugated to FlAsH in DCs. However, due to the scope of such endevour this can only be addressed as a separate follow up study. As for the preconjugation, we have tried and unfortunately failed to get it to work (Reviewer Figure 1). Therefore, we have shifted our focus to generating in-house peptide probes that are chemically conjugated to stable and bright fluorophore derivates. With that, we aim to circumvent the problems that the two-step FlAsH labeling poses.

Along those lines, do you have any way to quantify how many peptides you are detecting based on fluorescence? Being able to quantify the actual number of peptides would push the significance up.

We think two step procedure and background would pose challenges to such quantification in this study. although it would provide tremendous insight on the antigen-specific T cell- APC interactions in vivo, we think it should be performed using peptides chemically conjugated to fluorochromes with optimized signal to noise profiles.

In Figure 3D or 4 does the SA signal correlate with Flash signal on OT2 cells? Can you correlate Flash uptake with T cell activation, downstream of TCR, to validate peptide transfers?

To answer the reviewer’s question about FlAsH and SA correlation, we have revised the Figure 3d to show the correlation between OTII uptake of FlAsH, Streptavidin and MHCII. We also thank the reviewer for the suggestion on correlating FlAsH uptake with T cell activation and/or downstream of TCR activation. We have used proliferation and CD44 expressions as proxies of activation (Fig 2, 6). Nevertheless, we agree that the early events that correspond to the initiation of T-DC synapse and FlAsH uptake would be valuable to demonstrate the temporal relationship between peptide transfer and activation. Therefore, we have addressed this in the revised discussion.

**Author response image 4. sa3fig4:** FlAsH signal acquisition by antigen specific T cells is correlates with the OVA-biotin (SA) and MHCII uptake. DsRed+ splenic DCs were double-pulsed with 5 μM OVACACA and 5 μM OVA-biotin and adoptively transferred into CD45.1 recipients (2 × 106 cells) via footpad. Naïve e450-labeled OTII (1 × 106 cells) and e670-labeled polyclonal T cells (1 × 106 cells) were injected i.v. Popliteal lymph nodes were analyzed by flow cytometry. Overlaid histograms show the T cell levels of OVACACA (FlAsH) at 48 h post-transfer. Data are representative of three independent experiments with n=3 mice.

Minor:Figure 3F, 5D, and videos: Can you color-code polyclonal T cells a different color than magenta (possibly white or yellow), as they have the same look as the overlay regions ofOT2-DC interactions (Blue+red = magenta).

We apologize for the inconvenience about the color selection. We have had difficulty in assigning colors that are bright and distinct. Unfortunately, yellow and white have also been easily mixed up with the FlAsH signal inside red and blue cells respectively. We have now added yellow and white arrows to better point out the polyclonal vs. antigen specific cells in 3f and 5d.